# FROM NODES TO NETWORKS: EVOLVING RECURRENT NEURAL NETWORKS

## ABSTRACT

Gated recurrent networks such as those composed of Long Short-Term Memory (LSTM) nodes have recently been used to improve state of the art in many sequential processing tasks such as speech recognition and machine translation. However, the basic structure of the LSTM node is essentially the same as when it was first conceived 25 years ago. Recently, evolutionary and reinforcement learning mechanisms have been employed to create new variations of this structure. This paper proposes a new method, evolution of a tree-based encoding of the gated memory nodes, and shows that it makes it possible to explore new variations more effectively than other methods. The method discovers nodes with multiple recurrent paths and multiple memory cells, which lead to significant improvement in the standard language modeling benchmark task. Remarkably, this node did not perform well in another task, music modeling, but it was possible to evolve a different node that did, demonstrating that the approach discovers customized structure for each task. The paper also shows how the search process can be speeded up by training an LSTM network to estimate performance of candidate structures, and by encouraging exploration of novel solutions. Thus, evolutionary design of complex neural network structures promises to improve performance of deep learning architectures beyond human ability to do so.

## 1 INTRODUCTION

In many areas of engineering design, the systems have become so complex that humans can no longer optimize them, and instead, automated methods are needed. This has been true in VLSI design for a long time, but it has also become compelling in software engineering: The idea in "programming by optimization" is that humans should design only the framework and the details should be left for automated methods such as optimization (Hoos, 2012). Recently similar limitations have started to emerge in deep learning. The neural network architectures have grown so complex that humans can no longer optimize them; hyperparameters and even entire architectures are now optimized automatically through gradient descent (et al., 2016), Bayesian parameter optimization (Malkomes et al., 2015), reinforcement learning (Zoph & Le, 2016; Baker et al., 2016), and evolutionary computation (Miikkulainen et al., 2018; Real, 2017; Fernando, 2017). Improvements from such automated methods are significant: the structure of the network matters.

This paper shows that the same approach can be used to improve architectures that have been used essentially unchanged for decades. The case in point is the Long Short-Term Memory (LSTM) network Hochreiter & Schmidhuber (1997). It was originally proposed in 1992; with the vastly increased computational power, it has recently been shown a powerful approach for sequential tasks such as speech recognition, language understanding, language generation, and machine translation, in some cases improving performance 40% over traditional methods (Bahdanau et al., 2015). The basic LSTM structure has changed very little in this process, and thorough comparisons of variants concluded that there's little to be gained by modifying it further (Klaus et al., 2014; Jozefowicz et al., 2015).

However, very recent studies on metalearning methods such as neural architecture search and evolutionary optimization have shown that LSTM performance can be improved by complexifying it further (Zoph & Le, 2016; Miikkulainen et al., 2018). This paper develops a new method along these lines, recognizing that a large search space where significantly more complex node structures

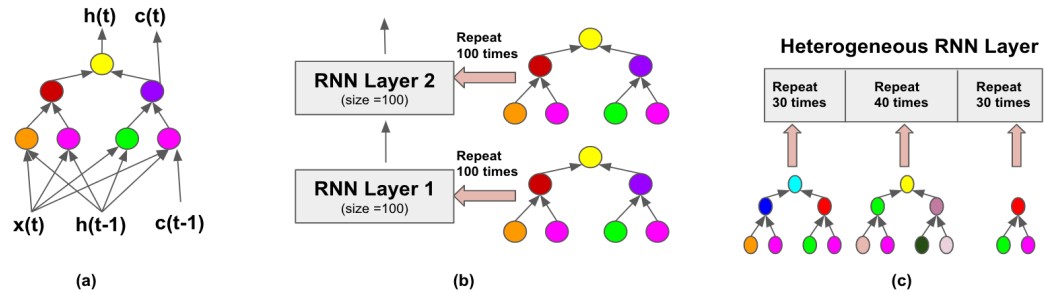

Figure 1: (a)Tree based representation of the recurrent node. Tree outputs $h(t)$ and $c(t)$ are fed as inputs in the next time step. (b) In standard recurrent network, the tree node is repeated several times to create each layer in a multi-layered network. Different node colors depict various element activations. (c) The heterogeneous layer consists of different types of recurrent nodes.

can be constructed could be beneficial. The method is based on a tree encoding of the node structure so that it can be efficiently searched using genetic programming. Indeed, the approach discovers significantly more complex structures than before, and they indeed perform significantly better: Performance in the standard language modeling benchmark, where the goal is to predict the next word in a large language corpus, is improved by 6 perplexity points over the standard LSTM (Zaremba et al., 2014), and 0.9 perplexity points over reinforcement-learning based neural architecture search (Zoph & Le, 2016).

These improvements are obtained by constructing a homogeneous layered network architecture from a single gated recurrent node design. A second innovation in this paper shows that further improvement can be obtained by constructing such networks from multiple different designs. As a first step, allocation of different kinds of LSTM nodes into slots in the network is shown to improve performance by another 0.5 perplexity points. This result suggests that further improvements are possible with more extensive network-level search.

A third contribution of this paper is to show that evolution of neural network architectures in general can be speeded up significantly by using an LSTM network to predict the performance of candidate neural networks. After training the candidate for a few epochs, such a Meta-LSTM network predicts what performance a fully trained network would have. That prediction can then be used as fitness for the candidate, speeding up evolution fourfold in these experiments. A fourth contribution is to encourage exploration by using an archive of already-explored areas. The effect is similar to that of novelty search, but does not require a separate novelty objective, simplifying the search.

Interestingly, when the recurrent node evolved for language modeling was applied to another task, music modeling, it did not perform well. However, it was possible to evolve another solution for that task that did. As a fifth contribution, the results in this paper demonstrate that it is not simply the added complexity in the nodes that matter, but that it is the right kind, i.e. complexity customized for each task.

Thus, evolutionary optimization of complex deep learning architectures is a promising approach that can yield significant improvements beyond human ability to do so.

## 2    BACKGROUND AND RELATED WORK

In recent years, LSTM-based recurrent networks have been used to achieve strong results in the supervised sequence learning problems such as in speech recognition [10] and machine translation (Bahdanau et al., 2015). Further techniques have been developed to improve performance of these models through ensembling (Zaremba et al., 2014), shared embeddings (Zilly et al., 2016) and dropouts (Gal, 2015).

In contrast, previous studies have shown that modifying the LSTM design itself did not provide any significant performance gains (Bayer et al., 2009; Cho et al., 2014; Jozefowicz et al., 2015). However, a recent paper from Zoph & Le (2016) showed that policy gradients can be used to train a LSTM network to find better LSTM designs. The network is rewarded based on the performance of

the designs it generates. While this approach can be used to create new designs that perform well, its exploration ability is limited (as described in more detail in Section 3.3). The setup detailed in Zoph & Le (2016) is used for comparison in this paper. In a subsequent paper Pham et al. (2018), the same policy gradient approach is used to discover new recurrent highway networks to achieve even better results.

Neuroevolution methods like NEAT (Stanley & Miikkulainen, 2002) are an alternative to policy gradient approaches, and have also been shown to be sucessful in the architecture search problem (Miikkulainen et al., 2018; Real, 2017). For instance, Cartesian genetic programming was recently used to achieve state of the art results in CIFAR-10 (Suganuma et al., 2017). Along similar lines, a tree based variant of genetic programming is used in this paper to evolve recurrent nodes. These trees can grow in structure and can be pruned as well, thus providing a flexible representation.

Novelty search is a particularly useful technique to increase exploration in evolutionary optimization (Lehman, 2012). Novelty is often cast as a secondary objective to be optimized. It allows searching in areas that do not yield immediate benefit in terms of fitness, but make it possible to discover stepping stones that can be combined to form better solutions later. This paper proposes an alternative approach: keeping an archive of areas already visited and exploited, achieving similar goals without additional objectives to optimize.

Most architecture search methods reduce compute time by evaluating individuals only after partial training (Suganuma et al., 2017; Real, 2017). This paper proposes a meta LSTM framework to predict final network performance based on partial training results.

These techniques are described in detail in the next section.

## 3 METHODS

Evolving recurrent neural networks is an interesting problem because it requires searching the architecture of both the node and the network. As shown by recent research (Zoph & Le, 2016) (Zilly et al., 2016), the recurrent node in itself can be considered a deep network. In this paper, Genetic Programming (GP) is used to evolve such node architectures. In the first experiment, the overall network architecture is fixed i.e. constructed by repeating a single evolved node to form a layer (Figure1(b)). In the second, it is evolved by combining several different types of nodes into a layer (Figure1(c)). In the future more complex coevolution approaches are also possible.

Evaluating the evolved node and network is costly. Training the network for 40 epochs takes two hours on a 1080 NVIDIA GPU. A sequence to sequence model called meta-LSTM is developed to speed up evaluation. Following sections describe these methods in detail.

### 3.1 GENETIC PROGRAMMING FOR RECURRENT NODES

As shown in Figure1(a), a recurrent node can be represented as a tree structure, and GP can therefore be used to evolve it. However, standard GP may not be sufficiently powerful to do it. In particular, it does not maintain sufficient diversity in the population. Similar to the GP-NEAT approach by Trujillo et al. Tujillo et al. (2015), it can be augmented with ideas from NEAT speciation.

A recurrent node usually has two types of outputs. The first, denoted by symbol $h$ in Figure1 (a), is the main recurrent output. The second, often denoted by $c$, is the native memory cell output. The $h$ value is weighted and fed to three locations: (1) to the higher layer of the network at the same time step, (2) to other nodes in the network at the next time step, and (3) to the node itself at the next time step. Before propagation, $h$ are combined with weighted activations from the previous layer, such as input word embeddings in language modeling, to generate eight node inputs (termed as base eight by Zoph & Le (2016)). In comparison, the standard LSTM node has four inputs (see Figure5(a)). The native memory cell output is fed back, without weighting, only to the node itself at the next time step. The connections within a recurrent cell are not trainable by backpropagation and they all carry a fixed weight of 1.0.

Thus, even without an explicit recurrent loop, the recurrent node can be represented as a tree. There are two type of elements in the tree: (1) linear activations with arity two (add, multiply), and (2) non-linear activations with arity one (tanh, sigmoid, relu, sin, cos).

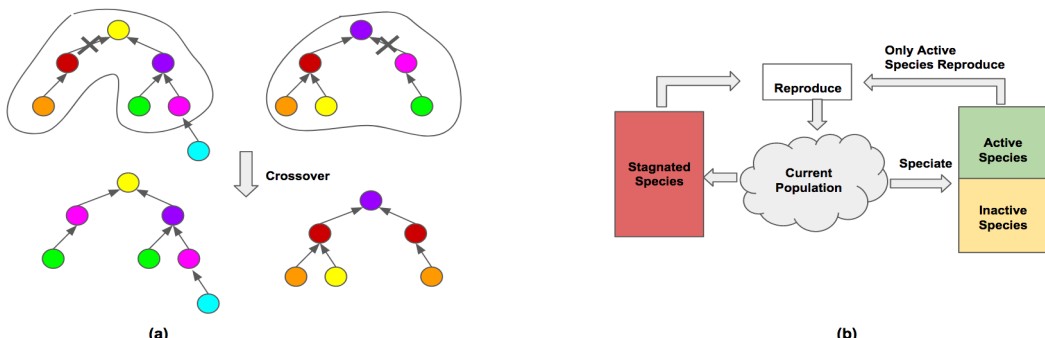

Figure 2: (a) Homologous crossover in GP - the two trees on the top look different but in-fact they are almost mirror images of each other. These two trees will therefore belong in the same species. The line drawn around the trees marks the homologous regions between the two. A crossover point is randomly selected and one point crossover is performed. The bottom two networks are the resultant offsprings. (b) An archive of stagnant species called Hall of Shame (shown in red) is built during evolution. This archive is looked up during reproduction, to make sure that newly formed offsprings do not belong to any of the stagnant species. At a time, only 10 species are actively evaluated (shown in green). This constraint ensures that active species get enough spawns to ensure a comprehensive search in its vicinity before it is added to the Hall of Shame. Offsprings that belong to new species are pushed into a inactive species list (shown in yellow) and are only moved to the active list whenever an active species moves to Hall of Shame.

There are three kind of mutation operations in the experiments: (1) Mutation to randomly replace an element with an element of the same type, (2) Mutation to randomly inserts a new branch at a random position in the tree. The subtree at the chosen position is used as child node of the newly created subtree. (3) Mutation to shrink the tree by choosing a branch randomly and replacing it with one of the branch's arguments (also randomly chosen).

One limitation of standard tree is that it can have only a single output: the root. This problem can be overcome by using a modified representation of a tree that consists of Modi outputs (Zhang & Zhang, 2004). In this approach, with some probability $p$ (termed modirate), non-root nodes can be connected to any of the possible outputs. A higher modi rate would lead to many sub-tree nodes connected to different outputs. A node is assigned modi (i.e. connected to memory cell outputs c or d) only if its sub-tree has a path from native memory cell inputs.

This representation allows searching for a wide range of recurrent node structures with GP.

## 3.2 SPECIATION AND CROSSOVER

One-point crossover is the most common type of crossover in GP. However, since it does not take into account the tree structure, it can often be destructive. An alternative approach, called homologous crossover (Francone et al., 1999), is designed to avoid this problem by crossing over the common regions in the tree. Similar tree structures in the population can be grouped into species, as is often done in NEAT (Tujillo et al., 2015). Speciation achieves two objectives: (1) it makes homologous crossover effective, since individuals within species are similar, and (2) it helps keep the population diverse, since selection is carried out separately in each species. A tree distance metric proposed by Tujillo et al. (2015) is used to determine how similar the trees are (see A.1 for detail).

In most GP implementations, there is a concept of the left and the right branch. A key extension in this paper is that the tree distance is computed by comparing trees after all possible tree rotations, i.e. swaps of the left and the right branch. Without such a comprehensive tree analysis, two trees that are mirror images of each other might end up into different species. This approach reduces the search space by not searching for redundant trees. It also ensures that crossover can be truly homologous Figure2 (a).

The structural mutations in GP, i.e. insert and shrink, can lead to recycling of the same strcuture across multiple generations. In order to avoid such repetitions, an archive called Hall of Shame

is maintained during evolution (Figure2(b)). This archive consists of individuals representative of stagnated species, i.e. regions in the architecture space that have already been discovered by evolution but are no longer actively searched. During reproduction, new offsprings are repeatedly mutated until they result in an individual that does not belong to Hall of Shame. Mutations that lead to Hall of Shame are not discarded, but instead used as stepping stones to generate better individuals. Such memory based evolution is similar to novelty search. However, unlike novelty search (Lehman, 2012), there is no additional fitness objective, simply an archive.

### 3.3 SEARCH SPACE: NODE

GP evolution of recurrent nodes starts with a simple fully connected tree. During the course of evolution, the tree size increases due to insert mutations and decreases due to shrink mutations. The maximum possible height of the tree is fixed at 15. However, there is no restriction on the maximum width of the tree.

The search space for the nodes is more varied and several orders of magnitude larger than in previous approaches. More specifically, the main differences from the state-of-the-art Neural Architecture Search (NAS) (Zoph & Le, 2016) are: (1) NAS searches for trees of fixed height 10 layers deep; GP searches for trees with height varying between six (the size of fully connected simple tree) and 15 (a constraint added to GP). (2) Unlike in NAS, different leaf elements can occur at varying depths in GP. (3) NAS adds several constraint to the tree structure. For example, a linear element in the tree is always followed by a non-linear element. GP prevents only consecutive non-linearities (they would cause loss of information since the connections within a cell are not weighted). (4) In NAS, inputs to the tree are used only once; in GP, the inputs can be used multiple times within a node.

Most gated recurrent node architectures consist of a single native memory cell (denoted by output $c$ in Figure1(a)). This memory cell is the main reason why LSTMs perform better than simple RNNs. One key innovation introduced in this paper is to allow multiple native memory cells within a node. The memory cell output is fed back as input in the next time step without any modification, i.e. this recurrent loop is essentially a skip connection. Adding another memory cell in the node therefore does not effect the number of trainable parameters: It only adds to the representational power of the node.

### 3.4 SEARCH SPACE: NETWORK

Standard recurrent networks consist of layers formed by repetition of a single type of node. However, the search for better recurrent nodes through evolution often results in solutions with similar task performance but very different structure. Forming a recurrent layer by combining such diverse node solutions is potentially a powerful idea, related to the idea of ensembling, where different models are combined together to solve a task better.

In this paper, such heterogenous recurrent networks are constructed by combining diverse evolved nodes into a layer (Figure1(c)). A candidate population is created that consists of top-performing evolved nodes that are structurally very different from other nodes. The structure difference is calculated using the tree distance formula detailed previously. Each heterogenous layer is constructed by selecting nodes randomly from the candidate population. Each node is repeated 20 times in a layer; thus, if the layer size is e.g. 100, it can consist of five different node types, each of cardinality 20.

The random search is an initial test of this idea. As described in Section 5, in the future the idea is to search for such heterogenous recurrent networks using a genetic algorithm as well.

### 3.5 META-LSTM FOR FITNESS PREDICTION

In both node and network architecture search, it takes about two hours to fully train a network until 40 epochs. With sufficient computing power it is possible to do it: for instance Zoph & Le (2016) used 800 GPUs for training multiple such solutions in parallel. However, if training time could be shortened, no matter what resources are available, those resources could be used better.

A common strategy for such situations is early stopping (Suganuma et al., 2017), i.e. selecting networks based on partial training. For example in case of recurrent networks, the training time

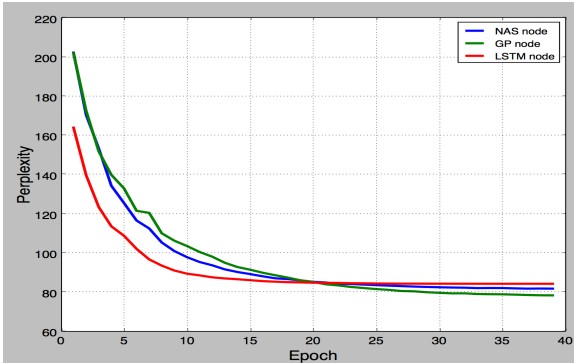

Figure 3: Learning curve comparison of LSTM node, NAS node and GP nodes. Y-axis is the validation perplexity (lower is better) and X-axis is the epoch number. Notice that LSTM node learns quicker than the other two initially but eventually settles at a larger perplexity value. This graph demonstrates that the strategy to determine network fitness using partial training (say based on epoch 10 validation perplexity) is faulty. A fitness predictor model like Meta-LSTM can overcome this problem.

would be cut down to one fourth if the best network could be picked based on the 10th epoch validation loss instead of 40th. Figure3 demonstrates that this is not a good strategy, however. Networks that train faster in the initial epochs often end up with a higher final loss.

To overcome costly evaluation and to speed up evolution, a Meta-LSTM framework for fitness prediction was developed. Meta-LSTM is a sequence to sequence model (Sutskever et al., 2014) that consists of an encoder RNN and a decoder RNN (see Figure4(a)). Validation perplexity of the first 10 epochs is provided as sequential input to the encoder, and the decoder is trained to predict the validation loss at epoch 40 (show figure). Training data for these models is generated by fully training sample networks (i.e. until 40 epochs). The loss is the mean absolute error percentage at epoch 40. This error measure is used instead of mean squared error because it is unaffected by the magnitude of perplexity (poor networks can have very large perplexity values that overwhelm MSE). The hyperparameter values of the Meta-LSTM were selected based on its performance in the validation dataset. The best configuration that achieved an error rate of 3% includes an ensemble of two seq2seq models: one with a decoder length of 30 and the other with a decoder length of 1 (figure).

Recent approaches to network performance prediction include Bayesian modeling (Klein et al. (2017)) and regression curve fitting (Baker et al., 2017). The learning curves for which the above methods are deployed are much simpler as compared to the learning curves of structures discovered by evolution (see Appendix). Note that Meta-LSTM is trained separately and only deployed for use during evolution. Thus, networks can be partially trained with a $4\times$ speedup, and assessed with near-equal accuracy as with full training.

## 4 EXPERIMENTS

Neural architectures were constructed for the language modeling task, using Meta-LSTM as the predictor of training performance. In the first experiment, homogeneous networks were constructed from single evolved nodes, and in the second, heterogeneous networks that consisted of multiple evolved nodes.

### 4.1 NATURAL LANGUAGE MODELING TASK

Experiments focused on the task of predicting the next word in the Penn Tree Bank corpus (PTB), a well-known benchmark for language modeling (Marcus et al., 1993). LSTM architectures in general tend to do well in this task, and improving them is difficult (Zaremba et al., 2014; Jozefowicz et al., 2015; Gal, 2015). The dataset consists of 929k training words, 73k validation words, and 82k test

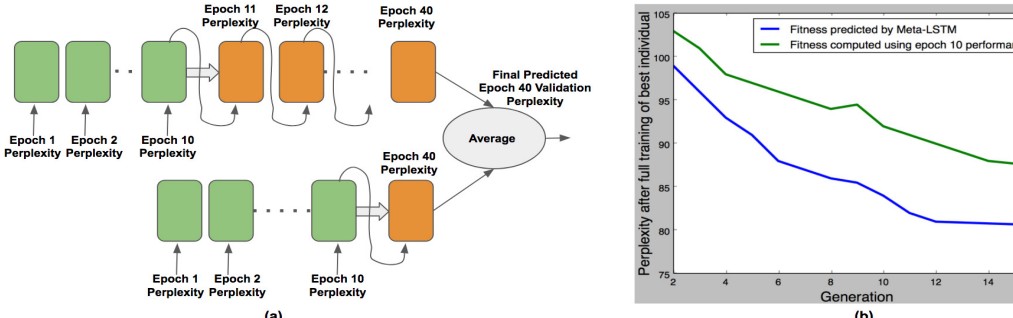

Figure 4: (a) Meta LSTM model: this is a sequence to sequence (seq2seq) model that takes the validation perplexity of the first 10 epochs as sequential input and predicts the validation perplexity at epoch 40. The green rectangles denote the encoder and the orange rectangles denote the decoder. Two variants of the model are averaged to generate one final prediction. In one variant (top), the decoder length is 30 and in the other variant (bottom), the decoder length is 1. (b) Meta LSTM performance: Two evolution experiments are conducted - one, where epoch 10 validation perplexity of the network is used as the fitness and second, where the value predicted by meta LSTM is used as the network fitness. After evolution has completed, the best individuals from each generation are picked and fully trained till epoch 40. For both the experiments, this graph plots the epoch 40 performance of the best network in a given generation. The plot shows that as evolution progresses, meta LSTM framework selects better individuals.

words, with a vocabulary of 10k words. During training, successive minibatches of size 20 are used to traverse the training set sequentially.

## 4.2 MUSIC MODELING TASK

Music consists of a sequence of notes that often exhibit temporal dependence. Predicting future notes based on the previous notes can therefore be treated as a sequence prediction problem. Similar to natural language, musical structure can be captured using a music language model (MLM). Just like natural language models form an important component of speech recognition systems, polyphonic music language model is an integral part of Automatic music transcription (AMT). AMT is defined as the problem of extracting a symbolic representation from music signals, usually in the form of a time-pitch representation called piano-roll, or in a MIDI-like representation.

MLM predicts the probability distribution of the notes in the next time step. Multiple notes can be turned on at a given time step for playing chords. The input is a piano-roll representation, in the form of an $88 \times T$ matrix $M$, where $T$ is the number of timesteps, and 88 corresponds to the number of keys on a piano, between MIDI notes A0 and C8. $M$ is binary, such that $M[p, t] = 1$ if and only if the pitch $p$ is active at timestep $t$. In particular, held notes and repeated notes are not differentiated. The output is of the same form, except it only has $T - 1$ timesteps (the first timestep cannot be predicted since there is no previous information).

The dataset piano-midi.de is used as the benchmark data. This dataset holds 307 pieces of classical piano music from various composers. It was made by manually editing the velocities and the tempo curve of quantized MIDI files in order to give them a natural interpretation and feeling ( Ycart & Benetos (2017)). MIDI files encode explicit timing, pitch, velocity and instrumental information of the musical score.

## 4.3 NETWORK TRAINING DETAILS

During evolution, each network has two layers of 540 units each, and is unrolled for 35 steps. The hidden states are initialized to zero; the final hidden states of the current minibatch are used as the initial hidden states of the subsequent minibatch. The dropout rate is 0.4 for feedforward connections and 0.15 for recurrent connections (Gal, 2015). The network weights have L2 penalty of 0.0001. The evolved networks are trained for 10 epochs with a learning rate of 1; after six epochs the learning rate is decreased by a factor of 0.9 after each epoch. The norm of the gradients (normalized by

minibatch size) is clipped at 10. Training a network for 10 epochs takes about 30 minutes on an NVIDIA 1080 GPU. The following experiments were conducted on 40 such GPUs.

The Meta-LSTM consists of two layers, 40 nodes each. To generate training data for it, 1000 samples from a preliminary node evolution experiment was obtained, representing a sampling of designs that evolution discovers. Each of these sample networks was trained for 40 epochs with the language modeling training set; the perplexity on the language modeling validation set was measured in the first 10 epochs, and at 40 epochs. The Meta-LSTM network was then trained to predict the perplexity at 40 epochs, given a sequence of perplexity during the first 10 epochs as input. A validation set of 500 further networks was used to decide when to stop training the Meta-LSTM, and its accuracy measured with another 500 networks.

In line with Meta-LSTM training, during evolution each candidate is trained for 10 epochs, and tested on the validation set at each epoch. The sequence of such validation perplexity values is fed into the trained meta-LSTM model to obtain its predicted perplexity at epoch 40; this prediction is then used as the fitness for that candidate. The individual with the best fitness after 30 generations is scaled to a larger network consisting of 740 nodes in each layer. This setting matches the 32 Million parameter configuration used by Zoph & Le (2016). A grid search over drop-out rates is carried out to fine-tune the model. Its performance after 180 epochs of training is reported as the final result (Table 1)

## 4.4 EXPERIMENT 1: EVOLUTION OF RECURRENT NODES

A population of size 100 was evolved for 30 generations with a crossover rate of 0.6, insert and shrink mutation probability of 0.6 and 0.3, respectively, and modi rate (i.e. the probability that a newly added node is connected to memory cell output) of 0.3. A compatibility threshold of 0.3 was used for speciation; species is marked stagnated and added to the Hall of Shame if the best fitness among its candidates does not improve in four generations. Each node is allowed to have three outputs: one main recurrent output ($h$) and two native memory cell outputs ($c$ and $d$).

The best evolved node is shown Figure5. The evolved node reuses inputs as well as utilize the extra memory cell pathways. As shown in Table 1, the evolved node (called GP Node evolution in the table) achieves a test performance of 68.2 for 20 Million parameter configuration on Penn Tree Bank. This is 2.8 perplexity points better than the test performance of the node discovered by NAS (Zoph(2016) in the table) in the same configuration. Evolved node also outperforms NAS in the 32 Million configuration (68.1 v/s. 66.5). Recent work has shown that sharing input and output embedding weight matrices of neural network language models improves performance (Press & Wolf, 2016). The experimental results obtained after including this method are marked as shared embeddings in Table 1.

It is also important to understand the impact of using meta LSTM in evolution. For this purpose, an additional evolution experiment was conducted, where each individual was assigned a fitness equal to its 10th epoch validation perplexity. As evolution progressed, in each generation, the best individual was trained fully till epoch 40. Similarly, the best individual from a evolution experiment with meta LSTM enabled was fully trained. The epoch 40 validation perplexity in these two cases has been plotted in Figure4(b). This figure demonstrates that individuals that are selected based upon meta LSTM prediction perform better than the ones selected using only partial training.

## 4.5 EXPERIMENT 2: HETEROGENEOUS RECURRENT NETWORKS

Top 10% of the population from 10 runs of Experiment 1 was collected into a pool 100 nodes. Out of these, 20 that were the most diverse, i.e. had the largest tree distance from the others, were selected for constructing heterogeneous layers (as shown in Figure1(c)). Nodes were chosen from this pool randomly to form 2000 such networks. Meta-LSTM was again used to speed up evaluation.

After hyperparameter tuning, the best network (for 25 Million parameter configuration )achieved a perplexity of 62.2, i.e. 0.8 better than the homogeneous network constructed from the best evolved node. This network is also 0.7 perplexity point better than the best NAS network double its size (54 Million parameters). Interestingly, best heterogeneous network was also found to be more robust to hyperparameter changes than the homogeneous network. This result suggests that diversity not only

Table 1: Single Model Perplexity on Test set of Penn Tree Bank. Node evolved using GP outperforms the node discovered by NAS (Zoph & Le, 2016) and Recurrent Highway Network (Zilly et al., 2016) in various configurations.

| Model | Parameters | Test Perplexity |
|---|---|---|
| Gal (2015) - Variational LSTM | 66M | 73.4 |
| Zoph & Le (2016) | 20M | 71.0 |
| GP Node Evolution | 20M | 68.2 |
| Zoph & Le (2016) | 32M | 68.1 |
| GP Node Evolution | 32M | 66.5 |
| Zilly et al. (2016) , shared embeddings | 24M | 66.0 |
| Zoph & Le (2016), shared embeddings | 25M | 64.0 |
| GP Evolution, shared embeddings | 25M | 63.0 |
| Heterogeneous, shared embeddings | 25M | 62.2 |
| Zoph & Le (2016), shared embeddings | 54M | 62.9 |

Table 2: F1 scores computed on Piano-Midi dataset. LSTM outperforms both the evolved node and NAS node for language, but not the node evolved specifically for music, demonstrating that the approach discovers solutions customized for the task.

| Model | F1 score |
|---|---|
| LSTM | 0.548 |
| Zoph & Le (2016) | 0.48 |
| GP Evolution (Language) | 0.49 |
| GP Evolution (Music) | 0.599 |

improves performance, but also adds flexibility to the internal representations. The heterogeneous network approach therefore forms a promising foundation for future work, as discussed next.

### 4.6 EXPERIMENT 3: MUSIC MODELING

The piano-midi.de dataset is divided into train (60%), test (20%) and validation (20%) sets. The music model consists of a single recurrent layer of width 128. The input and output layers are 88 wide each. The network is trained for 50 epochs with Adam at a learning rate of 0.01. The network is trained by minimizing cross entropy between the output of the network and the ground truth. For evaluation, F1 score is computed on the test data. F1 score is the harmonic mean of precision and recall (higher is better). Since the network is smaller, regularization is not required.

Note, this setup is similar to that of Ycart & Benetos (2017). The goal of this experiment is not to achieve state-of-the-art results but to perform apples-to-apples comparison between LSTM nodes and evolved nodes (discovered for language) in a new domain i.e. music.

In this transfer experiment, three networks were constructed: the first with LSTM nodes, the second with NAS nodes, and the third with evolved nodes. All the three networks were trained under the same setting as described in the previous section. The F1 score of each of the three models is shown in Table 2. LSTM nodes outperform both NAS and evolved nodes. This result is interesting because both NAS and evolved nodes significantly outperformed LSTM nodes in the language-modeling task. This result suggests that NAS and evolved nodes are custom solution for a specific domain, and do not necessarily transfer to other domains.

However, the framework developed for evolving recurrent nodes for natural language can be transferred to the music domain as well. The setup is the same i.e. at each generation a population of recurrent nodes represented as trees will be evaluated for their performance in the music domain. The validation performance of the network constructed from the respective tree node will be used as the node fitness. The performance measure of the network in music domain is the F1 score, therefore, it is used as the network fitness value.

The evolution parameters are the same as those used for language modeling. Meta-LSTM is not used for this evolution experiment because the run-time of each network is relatively small (¡ 600 seconds). The results from evolving custom node for music are shown in Table 2. The custom node (GP Evolution (Music)) achieves an improvement of five points in F1 score over LSTM (Figure 6). Thus, evolution was able to discover custom structure for the music modeling domain as welland it was different from structure in the language domain.

## 5 DISCUSSION AND FUTURE WORK

The experiments in this paper demonstrate how evolutionary optimization can discover improvements to designs that have been essentially unchanged for 25 years. Because it is a population-based method, it can harness more extensive exploration than other meta-learning techniques such as reinforcement learning, Bayesian parameter optimization, and gradient descent. It is therefore in a position to discover novel, innovative solutions that are difficult to develop by hand or through gradual improvement. Remarkably, the node that performed well in language modeling performed poorly in music modeling, but evolution was able to discover a different node that performed well in music. Apparently, the approach discovers regularities in each task and develops node structures that take advantage of them, thus customizing the nodes separately for each domain. Analyzing what those regularities are and how the structures encode them is an interesting direction of future work.

The GP-NEAT evolutionary search method in this paper is run in the same search space used by NAS (Zoph & Le, 2016), resulting in significant improvements. In a recent paper (Pham et al., 2018), the NAS search space was extended to include recurrent highway connections as well, improving the results further. An interesting direction of future work is thus to extend the GP-NEAT search space in a similar manner; similar improvements should result.

The current experiments focused on optimizing the structure of the gated recurrent nodes, cloning them into a fixed layered architecture to form the actual network. The simple approach of forming heterogeneous layers by choosing from a set of different nodes was shown to improve the networks further. A compelling next step is thus to evolve the network architecture as well, and further, coevolve it together with the LSTM nodes (Miikkulainen et al., 2018).

## 6 CONCLUSION

Evolutionary optimization of LSTM nodes can be used to discover new variants that perform significantly better than the original 25-year old design. The tree-based encoding and genetic programming approach makes it possible to explore larger design spaces efficiently, resulting in structures that are more complex and more powerful than those discovered by hand or through reinforcement-learning based neural architecture search. Further, these structures are customized to each specific domain. The approach can be further enhanced by optimizing the network level as well, in addition to the node structure, by training an LSTM network to estimate the final performance of candidates instead of having to train them fully, and by encouraging novelty through an archive. Evolutionary neural architecture search is therefore a promising approach to extending the abilities of deep learning networks to ever more challenging tasks.

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

## A  APPENDIX

### A.1  TREE DISTANCE

$$\delta(T_i, T_j) = \beta \frac{N_{i,j} - 2n_{S_{i,j}}}{N_{i,j} - 2} + (1 - \beta) \frac{D_{i,j} - 2d_{S_{i,j}}}{D_{i,j} - 2}, \tag{1}$$

where:

$n_{T_x}$ = number of nodes in GP tree $T_x$,

$d_{T_x}$ = depth of GP tree $T_x$,

$S_{i,j}$ = shared tree between $T_i$ and $T_j$,

$N_{i,j} = n_{T_i} + n_{T_j}$,

$D_{i,j} = d_{T_i} + d_{T_j}$,

$\beta \in [0, 1]$,

$\delta \in [0, 1]$.

On the right-hand side of Equation 1, the first term measures the difference with respect to size, while the second term measures the difference in depth. Thus, setting $\beta = 0.5$ gives an equal importance to size and depth. Two trees will have a distance of zero if their structure is the same (irrespective of the actual element types).

### A.2  EVOLVED SOLUTIONS

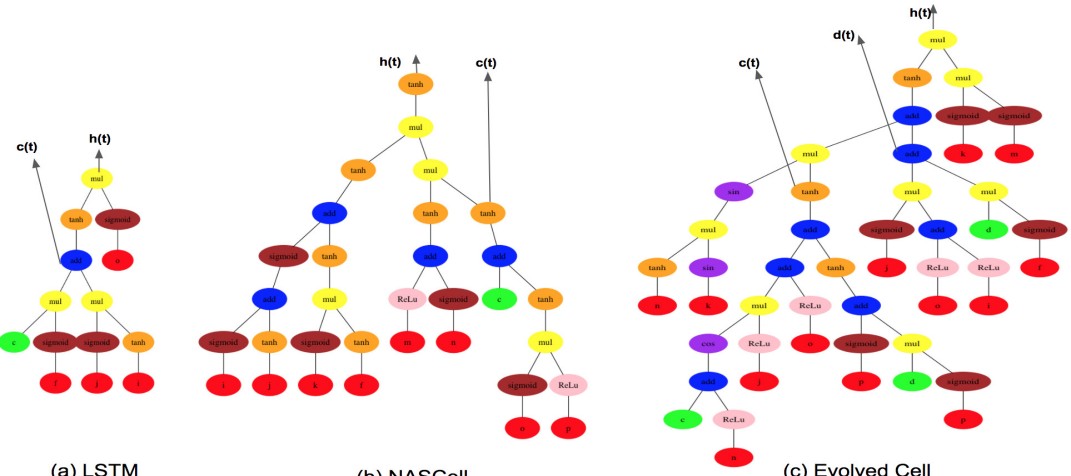

Figure 5: (a) Comparing Evolved recurrent node with NASCell and LSTM. The green input elements denote the native memory cell outputs from the previous time step $(c, d)$. The red colored inputs are formed after combining the node output from the previous time step $h(t-1)$ and the new input from the current time step $x(t)$. In all three solutions, the memory cell paths include relatively few non-linearities. The evolved node utilizes the extra memory cell in different parts of the node. GP evolution also reuses inputs unlike the NAS and LSTM solution. Evolved node also discovered LSTM like output gating.

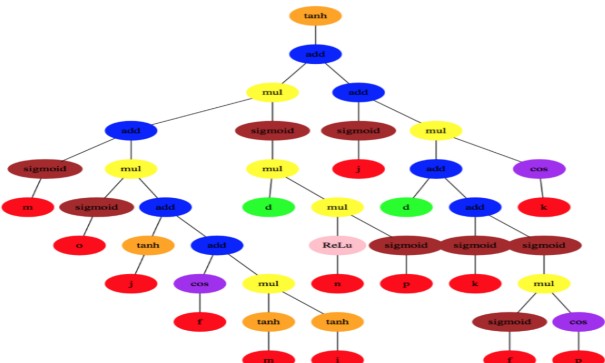

Figure 6: Evolved Node for Music. The node evolved to solve the music task is very different from the node for the natural language task. For example, this node only uses a single memory cell (green input $d$ in the figure) unlike the language node that used both $c$ and $d$. This results indicates that 'architecture does matter' and that custom evolved solution perform better than hand-designed ones.

