# OpenReview forum: "From Nodes to Networks: Evolving Recurrent Neural Networks"
_ICLR.cc/2019/Conference_

### Official Review · AnonReviewer2 · 2018-10-26
**Few contributions to architecture search, limited comparison to relevant work**

**Rating:** 4
**Confidence:** 4

**Review:**

The authors apply (tree-based) genetic programming (GP) to RNN search, or more specifically RNNs with memory cells, with the foremost example of this being the LSTM. GP provide a structured search that seems appropriate for designing NN modules, and has previously been applied successfully to evolving CNNs. However, the authors fail to mention that (tree-based) GP has been applied to evolving RNN topologies as far back as 2 decades ago, with even multiple cells in a single RNN unit [1]. The selection of more advanced techniques is good though - use of Modi for allowing multiple outputs, and neat-GP for more effective search (though a reference to the "hall of fame" [2] is lacking).

The authors claim that their method finds more complex, better performing structures than NAS, but allow their method to find architectures with more depth (max 15 vs. the max 10 of NAS), so this is an unfair comparison. It may be the case that GP scales better than the RL-based NAS method, but this is an unfair comparison as the max depth of NAS is not in principle limited to 10.

The second contribution of allowing heterogeneity in the layers of the network is rather minimal, but OK. Certainly, GP probably would have an advantage when searching at this level, as compared to other methods (like NAS). Performance prediction in architecture search has been done before, as noted by the authors (but see also [3]), so the particular form of training an LSTM on partial validation curves is also a minor contribution. Thirdly, concepts of archives have been in use for a long time [2], and the comparison to novelty search, which optimises for a hand-engineered novelty criteria, reaches beyond what is necessary. There are methods based on archives, such as MAP-Elites [4], which would make for a fairer comparison. However, I realise that novelty search is better known in the wider ML community, so from that perspective it is reasonable to keep this comparison in as well.

Finally, it is not surprising that GP applied to searching for an architecture for one task does not transfer well to another task - this is not specific to GP but ML methods in general, or more specifically any priors used and the training/testing scheme. That said, prior work has explicitly discussed problems with generalisation in GP [5].

[1] Esparcia-Alcazar, A. I., & Sharman, K. (1997). Evolving recurrent neural network architectures by genetic programming. Genetic Programming, 89-94.
[2] Rosin, C. D., & Belew, R. K. (1995, July). Methods for Competitive Co-Evolution: Finding Opponents Worth Beating. In ICGA (pp. 373-381).
[3] Zhou, Y., & Diamos, G. (2018). Neural Architect: A Multi-objective Neural Architecture Search with Performance Prediction. In SysML.
[4] Mouret, J. B., & Clune, J. (2015). Illuminating search spaces by mapping elites. arXiv preprint arXiv:1504.04909.
[5] Kushchu, I. (2002). An evaluation of evolutionary generalisation in genetic programming. Artificial Intelligence Review, 18(1), 3-14.

---

### Official Review · AnonReviewer3 · 2018-11-03
**Interesting but not enough**

**Rating:** 4
**Confidence:** 4

**Review:**

This paper explores evolutionary optimization for LSTM architecture search. To better explore the search space, authors used tree-based encoding and Genetic Programing (GP) with homologous crossover, tree distance metric, etc.  The search process is pretty simple and fast. However, there is a lack of experiments and analysis to show the effectiveness of the search algorithm and of the architecture founded by the approach.

Remarks:
The contents provided in this paper is not enough to be convinced that this is a better approach for RNN architecture search and for sequence modeling tasks.
This paper requires more comparisons and analysis.

Experiments on Penn Tree Bank
 - The dataset on both experiments are pretty small to know the effect of the new architecture they found. More experiments on larger datasets e.g., wikitext-2 will be needed.
 - In the paper "On the state of the art of evaluation in neural language models", Melis et al., 2018 reported improvement using classic LSTM over other variations of LSTM. They intensively compared the performance of classic LSTM, NAS, and RHN (Recurrent Highway Network) as authors did. Melis et al. reported LSTM (with depth 1) can already achieve a test perplexity of 59.6 with 10M parameters and 59.5 with 24M parameters.
- Could you analyze a new finding of the LSTM architecture compared to the classic LSTM and NAS? Figure 5 and 6 are not very clear how are their final architectures different and the important/useful nodes changes for different tasks?
- Recently, there are a number of architecture search algorithms introduced, but there is only one comparison in this direction (Zoph&Le16). It is important to compare this approach with other architecture search methods.

---

### Official Review · AnonReviewer1 · 2018-11-04
**An interesting idea but experiments and writeup need improvement.**

**Rating:** 5
**Confidence:** 4

**Review:**

A genetic algorithm is used to do an evolutionary architecture search to find better tree-like architectures with multiple memory cells and recurrent paths. To speed up search, an LSTM based seq2seq framework is also developed that can predict the final performance of the child model based on partial training results.

The algorithms and intuitions based on novelty search are interesting and there are improvements over baseline NAS model with the same architecture search space.

Although, the experiments are not compared against latest architectures and best results. For example on PTB, there are new architectures such as those created by ENAS that result in much lower perplexity than best reported in Table 1, for the same parameter size. While you have mentioned ENAS in the related work, the lack of a comparison makes it hard to evaluate the true benefit if this work compared with existing literature.

There is no clear abolition study for the Meta-LSTM idea. Figure 4 provides some insights but it'd be good if some experiments were done to show clear wins over baseline methods that do not employ performance prediction.

There are many typos and missing reference in the paper that needs to be fixed.

---

### Meta-Review · Area_Chair1 · 2018-12-14
**Work could be strengthened with additional experimental validation**

**Confidence:** 4
**Recommendation:** Reject

**Metareview:**

In this work, the authors explore using genetic programming to search over network architectures. The reviewers noted that the proposed approach is simple and fast. However, the reviewers expressed concerns about the experimental validation (e.g., experiments were conducted on small tasks; issues with comparisons (cf. feedback from Reviewer2)), and the fact that the method were not compared against various baseline methods related to architecture search.